# Ecosystems of Collaboration for Sustainability-Oriented Innovation: The Importance of Values in the Agri-Food Value-Chain

**José A. Gutiérrez [1,\*]**  **and Áine Macken-Walsh [2]**

1    Law Department, Universidad Santo Tomás, Cra. 82 #77BB-27, Robledo, Medellín 050034, Colombia
2    Teagasc, R93 XE12 Carlow, Ireland
\*    Correspondence: jose.danton@ustamed.edu.co; Tel.: +57-604-6040777

**Abstract:** There is growing recognition that sustainable innovation is not necessarily about new technologies, but about new or adapted organisational models, behaviours, and networks. How people engage in or with innovation is driven by values, but values differ across actor categories in agri-food value-chains. Understanding actors' values helps us to identify potential for collaborative innovation within agri-food value-chains, and to address potential barriers and obstacles. In the context of the Ploutos H2020 project, we conducted participatory focus group (FG) sessions at the EU level involving actors across the value-chain to brainstorm values, enablers, and hindrances in the process of sustainability-oriented innovation. Participants co-created stories showing scenarios within sustainability-oriented innovation where conflicts could occur between values and others where mutual values were created by multi-actor alliances. Based on a qualitative description of the data collected in these FGs, we identify a range of cultural and social values in decision-making and innovation processes, creating dilemmas and trade-offs, but also opportunities for sustainability-oriented innovation. A strong ecosystem of collaboration across the value-chain, based on relationships of shared interests and trust, is fundamental to innovation. We provide detailed insights regarding the use of participatory approaches to working with innovation actors to increase awareness of diversity in value systems and how it can be negotiated. Our findings are of particular interest to practice oriented scholars, practitioners, and innovation brokers working on the ground to further SOI.

**Keywords:** sustainability-oriented innovation; behavioural change; values; trust; agri-food value-chains

## 1. Introduction

As the long shadow of multiple economic, environmental, and social crises has loomed large on the horizon of contemporary debates on agri-food systems [1–4], there has been a renewed emphasis on sustainable solutions. Sustainable development is based on the idea that it is possible to integrate and balance the economic, social and environmental dimensions of development [5,6]. Yet, achieving this balance is exceedingly difficult [7,8]. In the agri-food sector:

> 'Economic issues include the incomes and livelihoods of producers and others involved in the network, employment, and local economic development, particularly in rural areas. Social issues include labour rights and the safety of workers, consumer health, food culture, and the accessibility, availability, and affordability of nutritious food (food security). Environmental impacts of food production, processing, packaging, distribution, and consumption, in turn, have to do with the use of resources and with pollution and damage to the soil, water, and air (including greenhouse gas emissions), biodiversity and ecosystems, and animal welfare' [9] (p. 65).

Sustainable innovation, in particular, has become a growing concept in articulating responses to these overlapping challenges. Sustainable innovation itself, however, is a topic of debate in the literature where a variety of definitions are employed. While an in-depth debate on the concept of sustainable innovation is beyond the scope of this paper, it is important to note that definitions of sustainability frequently emphasise technological aspects and the development of new products and/or commodities (e.g., [10,11]). This tendency has been described as a 'lock-in' of the agricultural sector into a technological paradigm [12], or even as a 'technical trap' [13], which prevents an improved understanding of the critical importance of societal structures and of people in change and innovation processes.

Notwithstanding the importance of technological developments, sustainable innovation is mostly about people, as recognised by a growing literature on the organisational and relational aspects to the success of sustainability-oriented innovation (SOI, i.e., innovations with the main purpose of bringing about outcomes that foster sustainability in the economic, social and environmental spheres) (for a discussion of this literature, see [14,15]). It is people who decide to engage (or not) with new technologies. People do and can (or do not or cannot) learn new ways of doing things and apply them. People make innovative organisational processes happen. People represent the critical component in any process of sustainable innovation, embracing it or resisting it. In this context, the importance of considering human behaviour to the sustainable innovation process becomes clear.

Behaviour, moreover, is not entirely random: it is structured around societal structures and people's values, which affect people's responses to innovation and inform decision-making processes [16–19]. As a working definition, we will refer to values as the properties we endow on things, actions, relations etc., making them important and relevant (or not) in our eyes, helping us to prioritise our actions, structuring our decision-making processes and forming the basis of human cognition (on values, see [20–22]).

This paper is a contribution to our understanding of how the values of various actors in the agri-food value-chain can impact the development of SOI. This research took place in the context of *Ploutos—Data Driven Sustainable Agri-Food Value-Chains*, an EU H2020 project with partners across 13 European countries. The project is very practical in orientation, hovering around 11 sustainability-oriented pilots in the agri-food sector, where SOI is designed and/or piloted, and tested. These pilots include, among many other initiatives, the development of sensors for precision agriculture, mobile applications for food donations, parametric insurance for farmers, etc.

The Ploutos project focuses on rebalancing the value chain for the agri-food system, along three innovation streams: behavioural innovation (i.e., behavioural adaptations necessary to adopt and/or co-create SOI across the agri-food ecosystem); Sustainable collaborative business model innovation (i.e., business models that support the re-balancing of the agri-food ecosystem to benefit the environment, society and the multiple actors involved in the value-chain); and data-driven technology innovation (i.e., re-using and extending technologies that support sustainability across the agri-food value-chain) (to learn more about the Ploutos project, please visit https://ploutos-h2020.eu/, accessed on 15 July 2022). In this paper we focus on one of these pillars, behavioural innovation, the importance of which is critical for the Ploutos project. In particular, we discuss one aspect of behavioural innovation: the values that shape and mould innovation processes, which will be discussed in the next section. The authors of this paper had a leading role in the work package directly relevant to the behavioural innovation pillar and a significant part of the preparatory work in this regard related to uncovering values in the SOI process to understand and increase awareness of enablers and hindrances in the project.

In this paper, we proceed by outlining the theoretical framework utilised in this paper, which applies Bourdieu's theory on the forms of capital [23] to understand values in SOI in the agri-food value-chain. We follow this by explaining the materials and methods on which we base our findings and discussion. We then present our analysis of data generated by four focus groups sessions (two groups were established, which met twice) conducted online in March–April 2021. In a discussion section we highlight the significance of diverse

values in SOI across the agri-food value-chain in the context of the existing literature, and the need for practical actions to respond to challenges. This informs our conclusions in relation to the importance of supporting strong ecosystems of collaboration to enhance the development of SOI.

## 2. Values and SOI

As discussed in the introduction, values are critical to shaping and moulding behaviour, which in turn is a fundamental aspect of SOI. We base our operational concept of values on Bourdieu's three forms of capital. Cultural capital in the institutionalised state refers to educational attainment (pride). Objectified cultural capital concerns the possession of cultural goods. Its embodied state refers to people's values, skills, knowledges and tastes. For Bourdieu, social capital is a network-based resource that is available in relationships and consequently accrues to individuals. He defines social capital as 'the aggregate of the actual or potential resources which are linked to the possession of a durable network of more or less institutionalized relationships of mutual acquaintance and recognition' [23] (p. 247). Economic capital refers to material assets that are 'immediately and directly convertible into money and may be institutionalized in the form of property rights' [23] (p. 242). Based on Bourdieu's model, we discuss three dimensions of values: *cultural values* (values aligned with pride and cultural distinctions of taste), *social values* (values of social relationships to actors), and *economic values* (the value of material wealth and monetary gain).

These values are not exclusionary of one another; people integrate them, with different emphases into their decision-making processes, in particular contexts, at specific times. Discussing how values inform (rather than dictate) decision-making, research shows how profit-making is insufficient as the sole motivator where enterprises are concerned [24]. Economic, social and cultural values are convertible across categories in various contexts, and they often overlap. How we balance them changes in time and place too. Although people are not always consistent, we argue that actors are more likely to engage with innovation if it resonates with their values, whether it is a desire for higher returns from their economic activities, intersecting with the availability of more time for family and social activities, and/or enhanced the sheer joy and pride of being a pioneer in one's field. People engaged in SOI may have diverging interests in the economic, social and environmental dimensions of sustainability, or they may engage in SOI for reasons other than an appreciation of sustainability itself [25,26]. However, in all cases, their decision-making process, regarding whether to engage or not in SOI, and how they engage with it, is driven by values.

Values are paramount in the agri-food sector too [27–30]. Most research on values in the agri-food sector has nonetheless focused on single or few actors within the value-chain. Yet, when we take a systems-based approach to thinking about sustainable agri-food chains, it is important to understand the differential values that different actors have in relation to the same issues. These values must be uncovered, exchanged and often mediated in order to chart conjoined processes across chains and systems where behavioural innovation is concerned. Values do not exist in isolation but are part of diverse networks, which are critical to agri-food systems [28,31].

Where agri-food ecosystems and food chains are concerned, we must understand how changes occur systemically; how 'social movements' occur within clusters or pockets of ecosystems/chains; and how they may connect across whole ecosystems/chains. Therefore, our attention is paid to how value systems are shared and differ within categories of actors (and the clusters/pockets they occupy), how values affect various actors' engagement with innovation, and the way in which synergies and trade-offs occur across categories (and parts of the system/chain) [29]. Through collaborations between actors from different sectors (clusters/pockets) of ecosystems/chains, multi-actor innovation can occur as a result of creatively combining different actors' values and knowledges, which can lead to powerful and transformative systemic change [32,33].

## 3. Materials and Methods

This research sought to uncover the values of different stakeholders and actors in the agri-food value-chain because of their importance to understand their perspective and behaviours (and therefore conflict, trade-offs and synergies) in relation to SOI. For this purpose, we used participatory focus groups (FGs), based on the EIP-Agri Focus Group (FG) model. The EIP-Agri FG model is based on the establishment of temporary groups of experts in a field who are diverse representatives of a sector in the agri-food value-chain; drawing from their expertise and experience, and facilitating them to discuss problems and opportunities, and propose solutions (to learn more about the EIP-Agri FGs, please visit https://ec.europa.eu/eip/agriculture/en/focus-groups, accessed on 12 July 2022). The FGs conducted were multi-actor in terms of approach, and we sought to include actors in agri-food ecosystems/chains across most of the countries where the Ploutos project has partners. Participants were from ten countries, including non-EU countries such as the UK and North Macedonia (see anonymised list of participants for details, Appendix A). We also sought to have a gender balance. Participants in the FGs included both partners within and outside of the Ploutos consortium. Face-to-face FGs were originally planned, but in the context of the COVID-19 pandemic, the FGs were held online, using Zoom and with the aid of participatory online tools, such as Klaxoon and Boords.

Recruitment of participants was achieved through the establishment of a recruitment working group (RWG), involving different partners within the Ploutos multi-actor consortium. The RWG held a recruitment workshop online, combining the use of Zoom and Klaxoon, a participatory online tool that allows simultaneous participatory interactions using a virtual whiteboard format (Figure 1). Through the multi-actor participants' expertise and knowledge, we brainstormed all the relevant sectors that we needed to represent in the FGs. After identifying all the relevant sectors, participants then brainstormed all the relevant actors within them. Thus, the whiteboard was populated with specific actors that our partners would recruit for the FGs. The RWG proceeded to contact specific people who corresponded with the sector/actor types identified in the workshop through a strategic snowballing approach.

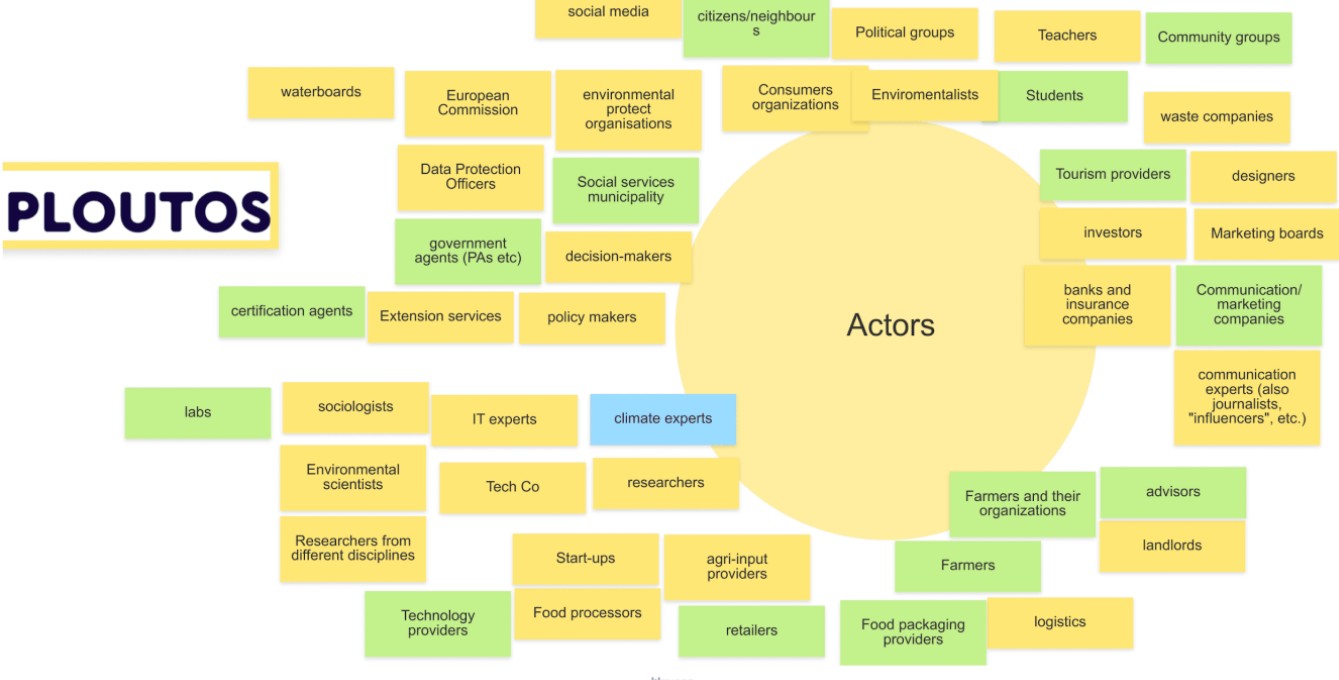

**Figure 1.** Actors identified through the RWG in Klaxoon.

The range of recruited participants for the FGs was balanced in terms of regions/countries and gender, although some sectors in the agri-food value-chain were better represented than others. Specifically, we found that consumers and community (development) actors were underrepresented -reflecting a tendency to underrepresentation of consumers in earlier similar research [25]. While most of the participants in the RWG identified consumers as an important sector for the FGs, few were capable of identifying participants representing consumers to recruit. As a result of this, for the purpose of consumers' representation, we relied heavily on one of the Ploutos consortium partners, a consumers' organisation. Participants representing the various sectors identified by the RWG (53 in total) were split into two groups (each with a similar membership profile), and each of the two groups met twice. The first session was called FGs1, and the second session was called FGs2. During FGs1, a vignette and brainstorming session was used; this took place for the first group on 17 March 2021, and for the second group, on 24 March 2021. FGs2 consisted of a storyboarding exercise, which took place for the first group on 31 March 2021 and for the second group on 7 April 2021. We will now proceed to explain the methods used in each of these FGs.

Ahead of FGs1, participants were requested to read and complete two vignettes -brief fictionalised stories that evoke real-life situations on innovation, conflict and collaboration (for use of vignettes in qualitative research, see [34–36]). The aim of this preparatory exercise was to sensitise participants to some of the diverse values of actors in the value-chain and to provoke thoughts on enablers/hindrances to innovation. Each vignette was part of a broader story, involving economic, social, and environmental dilemmas known to the facilitators, but unknown to the participants. All participants could see was the perspective of the two actors in the vignettes given to them, from of a total of five. The five actors in the vignettes were retailers, producers (food industry), producers (smallholders), authorities, and advisors, since these actors allowed us to create a story that could elicit discussions on values. We did not include a consumer actor in the vignette because of the aforementioned difficulty to recruit enough consumer representatives for the FGs, and we wanted participants to fill the vignette from the perspective of an actor closely resembling their perspective. Reducing all the possible actors to these five, allowed us, the researchers, to have in the vignettes a diverse group of actors of various sectors in the agri-food ecosystem while also using a workable number of actors in the stories and the vignettes. The participants had to provide an ending to the story, which was incomplete, prompted by the question 'what would you do if . . . ?', in which they pursued their own values and engaged with/negotiated with another actor's values in a SOI process (see Appendix D). The aim of this preparatory exercise was to provoke thought about SOI and different actors' roles in advance.

Once the FGs convened, with the use of Klaxoon, participants were requested to identify through brainstorming the *enablers and the hindrances* for sustainable innovation for each of the actors in the story (see Figure 2). Although in the vignettes there was no consumer because of the consumer representative deficit among participants, we decided to include the consumer actor perspective in the brainstorming session, by asking participants about what they believed would be a consumers' perspective in this story. This was facilitated by the perspective that, while not all of us are producers, advisors or authorities, we are all consumers. Participants separately brainstormed the enablers and hindrances identified in their particular vignettes through brainstorming and discussed them with other participants who had access to other vignettes, exchanging their respective perspectives, also thinking from the consumer perspective, and recording these perspectives by entering them on post-its on the Klaxoon whiteboard.

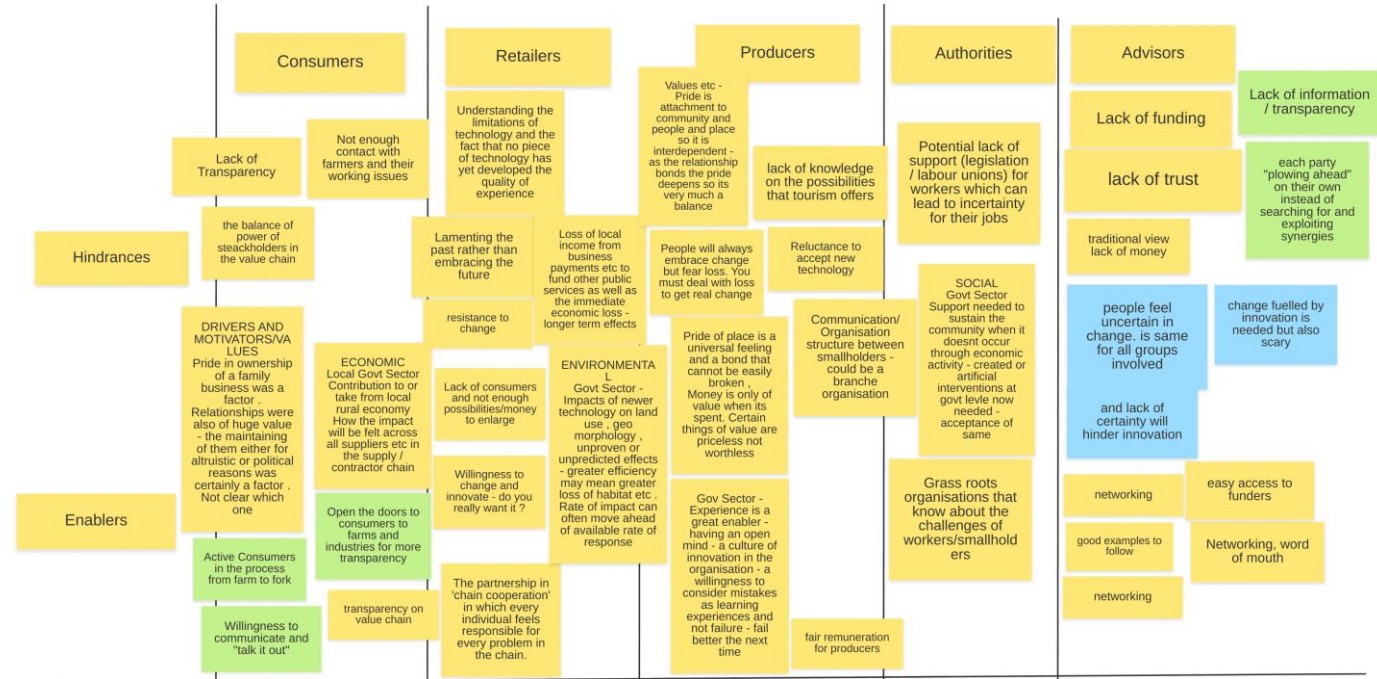

**Figure 2.** Enablers and hindrances for various actors in the vignettes as identified in FGs1.

Participants were then facilitated to identify *social, cultural and economic values* of each actor in their respective vignettes to help uncover the values of different actors across the value-chain, as reported on in the results section (see Figure 3). As a result, we developed a register of values that would help partners to negotiate values while engaged in SOI.

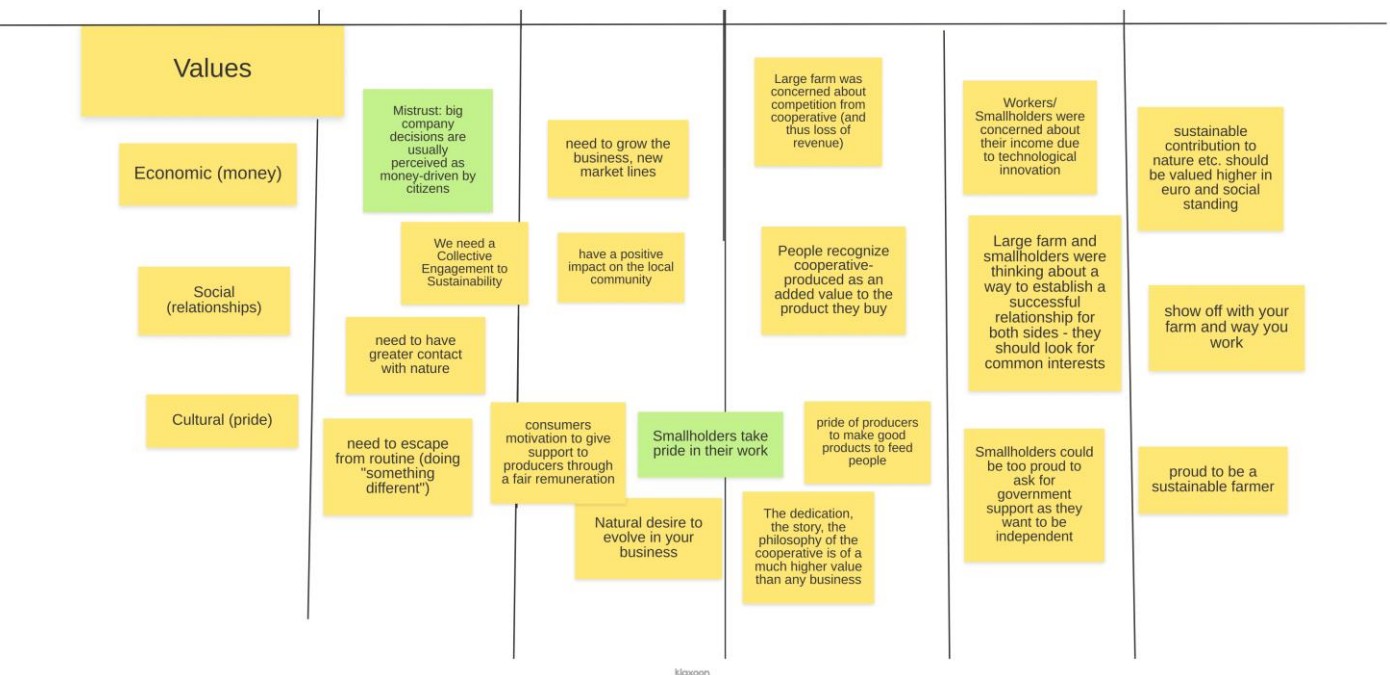

**Figure 3.** Values (economic, social, cultural) identified across the actors by participants in FGs1.

The subsequent FGs2 was aimed at further unpacking the values, knowledges, expertise and perspectives of participants in relation to conflict and collaboration and SOI in the agri-food value-chain. Participants engaged in a storyboarding exercise with the use of the

Boords platform, a software programme that supports the creation of storyboards online. With this exercise, we could explore, using a participatory approach, the perspectives and values of different actors within the agri-food value-chain and ecosystem, as well as their conflicts and potential for collaboration. Here, participants chose whatever actors they felt were needed in the story or they wanted to represent, so this expanded the list of five actors that were used in the vignettes. One researcher acted as note-taker while participants were requested to create their own story of sustainable-oriented innovation, each one of them representing a different character in the agri-food ecosystem (food retailer, farmer, tourism operator, etc.) (see Figure 4). This exercise facilitated participants to take a values-based approach and to discuss the challenges and potential solutions in building a story of SOI. This was achieved through the co-creation of a SOI-based story that was fictitious but represented different actors' values based on their own value-systems and experiences of interacting with others. Participants were asked, during the co-creation of the story, to refer to a register of values produced in the aftermath of the FGs1 vignettes for inspiration. We identified cross-themes and leverages across the values of actors in the ecosystems profiled in the stories, and we used these to structure our analysis.

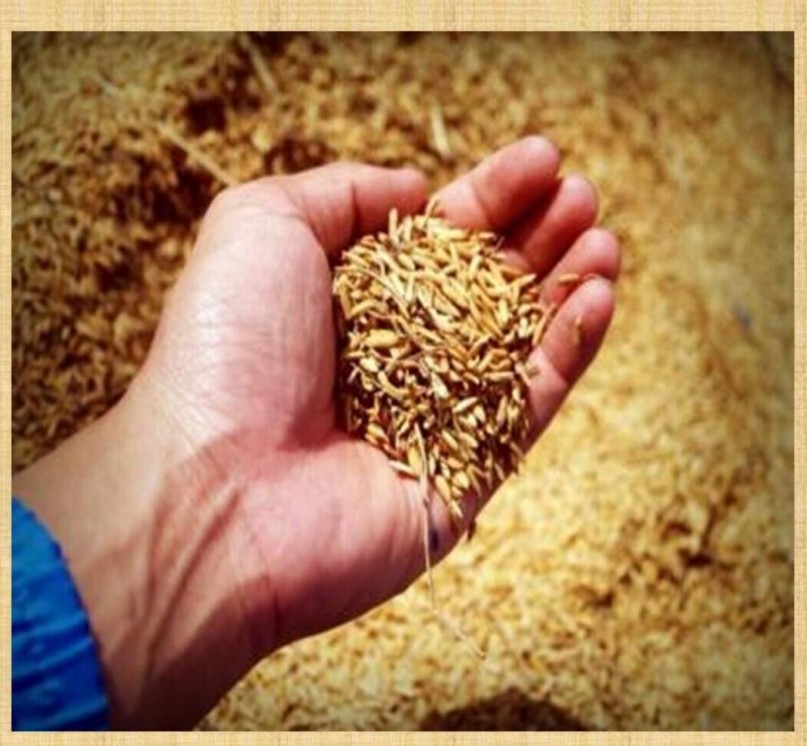

Peter wants to do the right thing and go organic, but demand for cheaper food – exacerbated by CAP- makes it difficult for him. He wants to plant wheat, but wheat is affected by global pricing. He needs support from authorities or someone to make this transition possible... Also he is aware that Anastasia, a local person involved in the food industry could produce the bread out of his wheat.

**Figure 4.** *Cont.*

**Figure 4.** Examples of images from the Storyboards created with the help of the Boords platform (each story consisted of 12 pages, with some text and one image).

## 4. Results

The data generated by the FGs were analysed following qualitative description, a methodology particularly suitable for research on human behaviour, which tends 'to draw from the general tenets of naturalistic inquiry' [37] (p. 337), 'entailing a commitment to studying a phenomenon in a manner as free of artifice as possible in the artifice-laden enterprise known as conducting research' [38] (p. 79). Analysis generated using qualitative description is particularly useful for presenting detailed cases that allow different readers' diverse interpretations of the cases for their own learnings and use. While we used Bourdieu to structure our theory of values, t the values were identified by participants themselves with as little interference as possible from the researchers. The results are presented according to the two FG sessions in which the two groups participated, FGs1 and FGs2. The former is based on the use of vignettes and brainstorming, and the latter is based on storyboarding. The results of both groups are combined for both FGs1 and FGs2, since both groups were comparable and they were only divided for practical purposes (i.e., to have a manageable number).

### 4.1. FGs1 (Vignettes and Brainstorming)

We will first present the economic, social and cultural values identified for these actors (Sections 4.1.1–4.1.3, respectively. See Appendix B). We will proceed by presenting the enablers and hindrances for sustainable innovation identified for each actor in the agri-food value-chain (Sections 4.1.4 and 4.1.5, respectively). Although there were distinctive clusters of values associated to different sectors/actors, there were also shared enablers and hindrances across sectors/actors.

#### 4.1.1. Economic Values

- Consumers: quality, precedence, transparency, price;
- Retailers: cost/profit, growth/diversification, and support for local producers;

- Producers: adequate income, collaboration, competitiveness;
- Authorities: adequate income, economic balance and fair trade;
- Advisors: value for money and quality.

4.1.2. Social Values

- Consumers: ethical consumption, health, tradition, and local values/cohesion;
- Retailers: collaboration and local support;
- Producers: inclusion, tradition and kin;
- Authorities: community commitment, family, guilt and shame about harming others;
- Advisors: change, cultural life and mutual learning.

4.1.3. Cultural Values

- Consumers: loyalty, trust, honesty, diversity, innovation;
- Retailers: innovation, prestige, tradition, diversity;
- Producers: pride, honour, prestige, tradition, loyalty, trust, independence/autonomy;
- Authorities: prestige, tradition and loyalty;
- Advisors: prestige and pride.

4.1.4. Enablers

- Consumers: described as mostly motivated by ethical and health-based decisions;
- Retailers: identified also as taking ethical considerations but mostly identifying market opportunities in sustainable products;
- Producers: mostly described as driven by the prestige of innovation, for their participation in wider networks, but also as motivated by new opportunities in the market;
- Authorities: enablers were described as deriving from organisational culture, opportunities for collaboration, and impact of development;
- Advisors: the enablers were described as a strong innovative ecosystem and good connections, mutual interests among stakeholders, desire to change and practical knowledge.

4.1.5. Hindrances

- Consumers: unrealistic expectations on price and the force of habit;
- Retailers: lack of market opportunities, cost/benefit;
- Producers: habit, lack of time/resources to spend on innovation, uncertainty;
- Authorities: unsuitable regulations, conflict of interests, distrust of people, lack of expertise or knowledge;
- Advisors: lack of trust, lack of transparency, and lack of skills.

*4.2. FGs2 (Storyboarding)*

The storyboarding exercise was used to create a multi-actor space to think about shared challenges and possible mutual sustainable solutions. The variety of values of these actors came to the fore (Appendix C), mediating between their knowledge and expertise. We will present both cross-themes that reflected the values of actors in the story (Section 4.2.1) and leverages (Section 4.2.2) which emerged in these stories.

4.2.1. Cross-Themes

- Ideas of fairness, transparency, honesty, equal opportunities and empathy underpin the importance given to trust across the agri-food value-chain. These values are linked by participants to the need for accurate information, effective communication, and for mutual understanding between actors.
- The ethical dimensions of production/consumption (animal welfare, environmental sustainability, supporting local produce, etc.) were very prominent for participants; although adequate income, good price and a margin of profit are important considerations, they didn't represent the sole drivers of decision-making.

- Among the challenges, lack of information or skills, an inadequate system of incentives, and lack of support (financial, legal, market) were identified by participants as threatening the viability of SOI.
- The solutions explored implied the resourceful use of existing opportunities; technological, organisational and behavioural innovations; and increased collaboration between actors across the agri-food value-chain.

### 4.2.2. Leverages

- When put in dialogue with one another, and in an environment fostering collaboration, complementarities between actors contributed to envisioning possible sustainable innovations to address common, real-life problems.
- Synergies between producers and consumers, consumers and retailers, the combination of local and external knowledge and expertise, provided a wealth of SOIs.
- It is particularly important to emphasise that although technological innovation played an important role in some stories, many of the solutions explored by participants required changes in collaboration patterns among actors, changes in behaviour or in organisational culture.

### 5. Discussion

A number of important considerations can be derived from these findings. Firstly, while economic factors, such as profit margins, the ability to derive an adequate income, and reasonable price, are important considerations for actors in the agri-food value-chain, they are not the only values present nor the most prominent when it comes to motivation or decision-making. This is consistent with many studies in the agri-food sector that that have used a Bourdieusian lens [39]. Contrary to rational-choice expectations, actors and stakeholders are moral agents who inhabit an ethical and cultural universe not reducible to profit calculations [27,40]. Their decision-making mechanisms are far more complex than pursuit of profit maximisation. Actors ponder options according to their values (economic, social and cultural), and ethical considerations; concern about the environment, peers, kin, others in the value-chain, etc. weigh heavily in decision-making and in determining actions (see [26]). Notwithstanding the fact that actors in the value-chain are not always fully consistent with their values in their respective courses of action across different contexts and scenarios [41], values nonetheless drive decision-making processes. Dilemmas in balancing (e.g. profit, price, demand; health, environment, etc.) thus arise in the decision-making processes of different actors in scenarios where SOI takes place. For instance, is it more important to have cheap food, healthy food, locally produced food, or food with a lower carbon footprint? Different actors, while acknowledging the importance of all of these attributes, will prioritise some over others, depending on circumstances, and striking a compromise that optimally balances these priorities requires (re)negotiating the values of these actors. It is important for both practitioners leading SOI and actors in the process to be sensitised to and aware of actors' diverse values in order to successfully negotiate the SOI process in a sustainable and innovative way, avoiding stand-offs and creating opportunities for the pursuit of mutual values. A fundamental understanding and awareness of different values is supportive of greater foresight in relation to why and how enablers and hindrances occur; and to proactively support enablers and avoid hindrances in the SOI process.

Beyond the level of individual actors and collaborative endeavours, features of the structures and systems in which SOI operates were also identified by participants. Participants identified social networks and organisational culture as deterministic of SOI. Innovation is not so much about each actor operating in isolation, but rather creating communities of innovation using a multi-actor approach or developing ecosystems which foster innovation. This emphasis on networks and the creation of strong ecosystems of innovation is consistent with previous research [32,33,42,43]. Within these ecosystems, shared values and organisational cultures can create a mixture of expectations that act as

an important enabler of innovation [14,28]. However, social capital within a network can act both as an enabler, but also as a hinderer of SOI, allowing networks, in some cases, to mobilise resources towards innovation, while in other cases having the opposite effect and effectively hindering innovative practices [44]. We identified force of habit, unrealistic expectations, and distrust towards certain actors all deeply embedded in social capital, as potential hinderers for SOI. This highlights the importance of identifying the potential for synergies as a conscious and self-reflexive process, such as by using the exercises employed in our FGs.

Trust was emphasised as critical in developing a strong culture of collaboration in networks and ecosystems (on the importance of trust in the agri-food chain see e.g., [31,44,45]. Trust brings together networks and communities of innovation, facilitating their joint endeavour and their confidence in its success [12,44,46]. However, trust is not a simple concept, having many layers and meanings. Newell & Swan [46], for instance, claim there is no one type of trust, but at least three types deriving from various modes of interaction: companion trust, springing out of a long-term interaction between actors; competence trust, based on the perceived or real expertise of a partner in the network; and commitment trust, based on mutual expectations and strong accountability mechanisms. Yet, power imbalances also create another challenge to the consolidation of trust within these ecosystems. Creating a culture of collaboration and trust across the whole value-chain requires, addressing and reconfiguring power imbalances within the agri-food value-chain [47] in order to create enabling spaces, particularly for disadvantaged or underrepresented groups.

Another important finding is that technological development did not arise as a magical solution to the various challenges faced by the agri-food value-chain actors during the storyboarding exercise. The best possible technology from the perspective of the technology developer is of little use if people do not engage with it, or, from the perspective of the end user, if it does not address their needs and desires. What's more, from a sustainability point of view, technologies, if not applied with an eye to social and environmental issues, can create new problems or unexpected impacts, such as power asymmetries and divides within the agri-food sector, widening already existing gaps [48]. Although technological development appeared in some of the storyboards, the majority hovered around organisational innovation and behavioural innovation. This emphasises the importance of the integrated approach adopted by Ploutos, which incorporates three pillars of innovation (behavioural, business model and technology) through concepts, such as technological co-creation, mutual value, and collaborative business models.

## 6. Conclusions

Understanding what will incentivise, motivate and drive actors to engage in and collaborate for SOI makes understanding their value systems (i.e., what they value) critically important. The different value systems that actors have can produce both frictions and potential for mutual collaborations between actors across chains and wider systems. We created a register showing different actors' values and we used the storyboarding tool to generate scenarios of collaborative SOI, in an approach to practically assist multi-actor SOI. Though attributing a standardised set of values to groups of people or sectors within a value-chain or ecosystem is useful for generating awareness of actors' different values that must be met in the SOI process, it is also the case that homogeneity in values within actor categories is improbable in practice [20,49]. Previous research has demonstrated how farmers with similar socio-economic backgrounds can have diverging attitudes and behaviours in relation to innovation [42,43,45]. Similarly, in research in relation to consumers' habits [50,51] and retailers [52,53], Sánchez-González et al. [30] have demonstrated that 'all-things-equal' scenarios do not guarantee the same values towards sustainability. That means that the same actors in similar conditions can still make divergent decisions. This makes clear the need for SOI processes to implement the participatory approaches followed in this paper throughout the SOI process, incrementally charting not only actors' values at the outset of initiatives, but charting changes in value systems and new opportunities for

pursuing mutual value. Understanding the (changing) variability of values within those groups in each individual multi-actor context, and how this affects the SOI process, remains an important challenge to be addressed.

Although all individuals will differ in terms of emphasis in their values, we are interested in the ways in which contexts/habitats that people share lead to them having some shared values. The shared position that any group of actors has in an agri-food ecosystem/an agri-food value-chain can shape a collection of values that they share/collectively hold. Sociology, through the study of group behaviour, can offer guiding insights to cohorts of actors, how the values of actors intersect, and their significant differences. Understanding shared values within groups can help us explain why and how innovations are engaged with or not by various actors, and the extent to which innovations are to different extents enthusiastically embraced or not (and further innovated or not). Further insights from experiences in the field where actors aim to collaborate for SOI stand to contribute to knowledge and conventional wisdom for supporting SOI, in particular, negotiating diverse values in forging sustainable innovation pathways.

**Author Contributions:** Writing—original draft, J.A.G.; Writing—review & editing, Á.M.-W. All authors have read and agreed to the published version of the manuscript.

**Funding:** This research was funded by the European Union, through the Horizon 2020 research and innovation programme, under grant agreement 101000594.

**Institutional Review Board Statement:** This study followed the ethic requirements of the Ploutos H2020 project, which were established in accordance with the European Commission, under grant agreement 101000594.

**Informed Consent Statement:** Informed consent was obtained from all subjects involved in the study.

**Data Availability Statement:** Data may be requested from the corresponding author. Requests will be evaluated on a case by case basis and will be subject to GDPR and other privacy constraints.

**Acknowledgments:** The authors would like to acknowledge Caroline van der Weerdt, Dianne van Hemert, Anita Naughton, Christopher Brewster, Nikolaos Marianos, Jack McCarthy, and Martin McNamara for insights and feedback provided on the Focus Groups and deliverables on which this paper is based.

**Conflicts of Interest:** The authors declare no conflict of interest. The funders had no role in the design of the study; in the collection, analyses, or interpretation of data; in the writing of the manuscript, or in the decision to publish the results.

## Appendix A

**Table A1.** Table of Participants to the FGs.

| Actor | Country | Gender |
|---|---|---|
| Consumers | Greece | Male |
| Food Industry | Cyprus | Male |
| | Cyprus | Male |
| | UK | Female |
| | UK | Male |
| | France | Male |
| | Italy | Female |
| | Spain | Female |
| | Greece | Male |

**Table A1.** *Cont.*

| Actor | Country | Gender |
|---|---|---|
| Advisor | Italy | Female |
| | Cyprus | Male |
| | Greece | Female |
| | Greece | Male |
| | North Macedonia | Male |
| Farmer | Spain | Female |
| | Ireland | Male |
| | France | Male |
| Retail | Spain | Male |
| | Spain | Male |
| | Netherlands | Male |
| Tourism | Ireland | Male |
| | Ireland | Female |
| | Spain | Female |
| | Spain | Female |
| | Spain | Male |
| Research | Netherlands | Female |
| | Spain | Male |
| | Spain | Female |
| | Belgium | Female |
| | Spain | Female |
| | Spain | Female |
| | Spain | Male |
| | Cyprus | Male |
| | Netherlands | Male |
| | Netherlands | Male |
| | North Macedonia | Female |
| | Ireland | Female |
| | Spain | Female |
| Development Sector | Spain | Female |
| | Spain | Female |
| | Spain | Female |
| | Spain | Male |
| Government | Netherlands | Male |
| | Netherlands | Female |
| | Cyprus | Female |
| | Ireland | Female |
| | Cyprus | Male |
| | Netherlands | Female |
| | Netherlands | Male |
| | Ireland | Male |

**Table A1.** *Cont.*

| Actor | Country | Gender |
|---|---|---|
| | Ireland | Male |
| Finance Sector | Spain | Female |
| | Netherlands | Male |

## Appendix B

**Table A2.** Collated Data with the Values Identified by Participants in FGs1 during the Brainstorming Session in Klaxoon.

| Actor | Economic Values | Social Values | Cultural Values |
|---|---|---|---|
| Consumers | • Bigger range to choose from<br>• Distrust (in companies)<br>• Quality of production<br>• Fair price/trade<br>• Transparency<br>• Support the local<br>• Reduce costs<br>• Standard of life | • Animal welfare<br>• Tradition<br>• Local products (added value)<br>• Local community cohesion<br>• Community vibrancy/viability<br>• Cultural values<br>• Safer environment & Contact with nature<br>• Healthy relations<br>• Collective engagement | • Loyalty (community, family)<br>• Trust<br>• Do something different (innovate)<br>• Diversity<br>• Honesty<br>• Individualism/Collectivism |
| Retailers | • Diversification<br>• Growth<br>• Support the local<br>• Fair price/trade<br>• Reduce costs<br>• Profit | • Community vibrancy/viability<br>• Commitment to local<br>• Collaboration<br>• Local products (added value) | • Evolve (innovate)<br>• Reluctance to change<br>• Prestige (Quality, pioneers)<br>• Diversity |
| Producers | • Sustainable family income<br>• Market competition<br>• Cross-sectoral cooperation<br>• Adequate income<br>• Profit<br>• Reduce costs<br>• Collaboration<br>• Standard of life<br>• Quality of production<br>• Competitiveness | • Inclusion<br>• Gender Balance<br>• Family<br>• Collaboration<br>• Tradition<br>• Cultural values | • Prestige (Quality, pioneers)<br>• Pride<br>• Honour<br>• Independence/autonomy<br>• Dedication<br>• Expectations and standing in community<br>• Individualism/Collectivism<br>• Tradition (improve through innovation)<br>• Reluctance to change<br>• Loyalty (community, family)<br>• Trust |
| Authorities | • Economic balance<br>• Fair price/trade<br>• Adequate income | • Guilt and shame<br>• Family<br>• Community vibrancy/viability<br>• Commitment to local | • Expectations and standing in community<br>• Individualism/Collectivism<br>• Tradition (improve through innovation)<br>• Loyalty (community, family) |
| Advisors | • Reduce costs<br>• Collaboration<br>• Quality of production | • Change<br>• Cultural life<br>• Mutual learning | • Prestige (Quality, pioneers)<br>• Pride<br>• Individualism/Collectivism |

## Appendix C

**Table A3.** Values Identified in the Storyboarding Exercise.

| Actor | Values |
|-------|--------|
| Producer | • Ethical considerations (animal welfare, environmentally friendly) <br> • Adequate income <br> • Mutual aid <br> • Education |
| Consumer | • Ethical considerations (animal welfare, environmentally-friendly, fair trade) <br> • Distrust of companies <br> • Affordable price <br> • Transparency <br> • Education <br> • Information <br> • Empathy (identification) |
| Retailer | • Diversity <br> • Profit <br> • Support local <br> • Ethical considerations (environmentally-friendly, fair trade) <br> • Honesty (branding) <br> • Information <br> • Regulations compliance |
| Tourism Operator | • Profit <br> • Support local <br> • Equal opportunities |
| Authorities | • Reward <br> • Support <br> • Regulations <br> • Participation <br> • Trust <br> • Pride <br> • Success |
| Advisors | • Know-how <br> • Transparency <br> • Compliance |
| Researchers | • Understanding difference |
| Marketing consultant | • Change <br> • Engagement <br> • Communication |
| Environmentalist | • Support <br> • Transparency <br> • Robust regulations |
| Innovation broker | • Communication |
| Banker | • Being alert for opportunities <br> • Patience <br> • Profit |
| Social media influencer | • Communication |

## Appendix D

**Box A1.** Vignettes Circulated to Participants Ahead of FGs1.

---

**WP2 F.G.1. Sustainable behavioural innovation—what would you do if . . . ?**
Name:
Organisation:
Country:
Dear participant,
Thank you very much for accepting to participate on a voluntary basis in these workshops [i.e., FGs] aiming to explore the behavioural dimensions of sustainable innovation. In this first exercise, you will be requested to complete two fictional and inter-related stories. One of the stories involves a protagonist that is close to your own area of expertise; the other story, involves a protagonist who is not within your area of expertise. Completing both stories will require that you think of what decisions you, but also other stakeholders in the value-chain, would make in such a situation.
Try to tell us what you think could happen and how this story could end. As you complete these stories (no more than 600 words each) try to think of the consequences of your decisions on other stakeholders, and also of the inducers and barriers which you would face.
**Vignette 1: The Mahon family business.** The Mahon's have run their fruit business for generations. When they started their business, most of the farmers in the region were smallholders engaged in subsistence farming; but the Mahons had the vision, the land and the right connections to take advantage of the benign conditions for large-scale fruit production. They were much beloved by the locals, since they represented not only something to be proud of in this remote region, but also they provided much needed seasonal work for the smallholders and were the backbone over which a dynamic trade rested.
But times changed, and their business came under much difficulties to conform to environmental regulations. Although there was little concern about the extensive use of pesticides in the past, now this was a pressing issue. Looking for solutions, the Mahons were offered a technological solution—a system of cameras which scan the leaves, identify the pests in their bud, and therefore offered the opportunity to apply pesticides selectively and strategically. This technology is effective, much less labour intensive, and greener. However, it is cost-effective only in the long term, as the costs for acquiring this technology is very high.
The Mahons are under pressure to comply with the regulations and need to come up with a solution to stay on business and to conform to the norms . . .
What would you do if you were the Mahons? What sustainable options could they take?
**Vignette 2: Sabina, the smallholder.** Sabina is a smallholder. She grows vegetables in a few hectares, has a few cows which she milks, and gets a small income from selling some of her produce to local traders. She and her husband have supplemented their income with seasonal work at the Mahon's farm. They are a family that for generations have run a successful fruit business. They employ hundreds of local labourers on a seasonal basis, and all of these families depend on them for their subsistence. Locals labour for them in the harvest season, but another big job on which many depend is the control of plagues and the use of pesticides. Word has spread around that the Mahons will be installing a new camera system which will require much less labour for controlling plagues and for fumigations, since the latter will be strategic and focalised.
Loss of that extra-income would be disastrous for Sabina. Her husband is one of the few lucky ones who had been offered a job after the new system is in place. However, the family relies on Sabina's income too. There are many others in a similar situation and they have come together to decide what to do in relation to this situation.
What would you do if you were Sabina? What sustainable options could she take?
**Vignette 3: Martin, advisor.** Martin is an agronomist who has worked extensively in advisory roles across the whole country. He comes for a farmers' family. Everyone in his family have worked on farming, but his generation, after attending college, decided to work as professionals and eventually the family lands were sold. He knows very well the struggles of many farmer families to make ends meet –from his own experience! He believes that technology can play a vital role to increase the productivity of farms, as well as to reduce costs. He has developed good business relations with a particular brand who are now offering a new technological package to deal with pests: a system of cameras that scan the fields and/or grow houses, spotting the plagues and providing data for strategic and discrete fumigation. This is a costsaving technology in the long run, despite its initial high costs. It is also more environmentally friendly than the indiscriminate use of pesticides and it also requires less labourers.
He is approached by the Mahons, a family of medium to large scale fruit producers, who are coming under increasing pressure to conform to environmental regulations. They are particularly concerned about the extensive use of pesticides and this is an area in which they would like to improve. He offers them the new package of cameras, the cost of which is a bit out of reach for the Mahons. He is aware there may be other environmentally, cost-effective and sound solutions, but he is also committed to promote this particular brand because of the long-term relationship with them. He also knows that their technologies are efficient and good quality.
What would you do if you were Martin? What sustainable options could he take?

---

**Vignette 4: Marianne and Thomas, retailers.** Marianne and Thomas own a retailer and trade firm in a small rural village which produces mostly fruits, although small farmers produce small quantities of dairy products as well—cheese, butter and milk. Fruit production is mostly linked to the Mahons, a family who has substantial land and have run their business for generations. They have been traditionally good partners to Marianne and Thomas, selling their high quality fruits in their market and across the country. They are the main traders in the region and everybody goes to their supermarket to do the shopping.

Recently, the Mahons acquired a new system of cameras and fumigation, which requires far less labour than the previous extensive use of pesticides. Although they are very happy with the new cleaner technology, this has a negative impact over their business: since seasonal work was crucial for smallholders, now, that they are no longer needed in the same quantity as before for fumigation, there has been a pronounced drop in the sales at their supermarket. But they currently face a dilemma—some smallholders have recently formed a cooperative to put together their resources, and although some of their products do not compete with the Mahons' (like dairy products), they have started producing fruits too. This makes them a direct competition for the Mahons family business. Marianne and Thomas can only buy so much fruit from each source –and it is better for them to reach an agreement with only one of them, whether the Mahons or the cooperative.

What would you do if you were Marianne and Thomas? What sustainable options could they take?

**Vignette 5: Rachel Mahon, councillor.** Rachel Mahon is a councillor in a rural region, who is well-known by the community. She is a young member of the Mahon family, the biggest local business and land-owners, a family of fruit producers who have run their business for generations. Their family have traditionally provided much needed seasonal work for local smallholders. However, her relatives have acquired new technologies which make a lot of the traditional work provided by the family redundant. She is aware that being a member of the family, she is likely to become all of a sudden very unpopular. However, she is genuinely concerned about many families that will suffer because of this decision. In her work as councillor she has visited many of these families and has worked on a number of initiatives to improve the situation of these farmers.

She is approached by some of the smallholders who have recently set up a cooperative looking for institutional support for their community-based business. This causes much distress for her, since she has an obligation with the community, while at the same time, she feels that she is supporting her family's competition.

How can she find a solution which works both for the long-standing business in the region (her family's) but also for her own constituents who formed the cooperative? What would you do if you were Rachel? What sustainable options could she take?

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
