# Peer review of "Ecosystems of Collaboration for Sustainability-Oriented Innovation: The Importance of Values in the Agri-Food Value-Chain"

_sustainability, doi:10.3390/su141811205_

Round 1
Reviewer 1 Report
Article
Ecosystems of Collaboration for Sustainability-Oriented Inno- vation: The Importance of Values in the Agri-Food Value- Chain.
The authors propose an interesting study under the auspices of the Ploutos H2020 project, which can be consulted on the project website. Sustainability, innovation, and how this is seen through the agri-food supply chain.
Using qualitative analysis and focus groups among supply chain actors, the authors analyze which values can facilitate SOI (Sustainability Oriented Innovation). The paper shows how values vary along the supply chain, and also shows that economic profit is not the only driver of stakeholders' decisions. The research encourages the search for synergies and the deepening of relationships and networks between supply chain actors. It is therefore a very interesting study and research, however the paper presented is, at times, difficult to understand for an outsider. Therefore, the authors should improve several elements of their study, some of which are presented below. The formal elements seem simple to address, but the really important ones are those that concern the methodology and the results. Best of luck and encouragement in your work
0) Abstract: we conducted four participatory focus groups (FGs). It is not clear throughout the article whether there are 2 or 4 focus groups, or if they are workshops as stated in Appendix A.
1) Introduction, first paragraph. Misquote: …. on agri-food systems [1][2][3][4][5][6], see author’s guide. This error is repeated throughout the article.
2) Introduction, first paragraph. You have to put a semicolon and hyphens in the next two paragraphs which also have a different line passage that should not be. ….Yet, achieving this balance is exceedingly difficult [9][10]. In the agri-food sector:
3) Pag 2, first paragraph…. water, and air (including greenhouse gas emissions), biodiversity and ecosystems, and animal welfare [11](p. 65). is not enclosed in quotation marks and is a verbatim citation.
4) Pag 2, second paragraph. It is necessary to know what we mean by SOI in this article, you must define it.
5) Pag 2, third paragraph. Sustainability-Oriented Innovation (SOI), Are you sure that having initialed the abstract, you have to initial it here as well? Check the author's guide.
6) Pag 2, third paragraph. …….(for a discussion of this literature, see [16]; see also [17]). Does not fit well, it looks better to see [16-17].
7) Pag2. 4th paragraph…. responses to innovation [18]. This quote is very old, there are in others from '2020, for example Bouman....
8) Pag2. 4th paragraph…. of human cognition (on values, see [18,19,22–24]. Misquoted parenthesis that opens but does not close.
9) Pag2. 5th paragraph…. The Ploutos project focuses on rebalancing the value chain for the agri-food system, along three innovation streams: Behavioural Innovation; Sustainable Collaborative Business Model Innovation and Data-driven Technology Innovation. Here we are focusing on one of these pillars, behavioural innovation, whose importance is understood as critical for the project. But in the whole article, Behavioural Innovation is not mentioned again, only SOI is mentioned. The authors should define what is the objective of their article because it is puzzling for the reader.
10) Pag3. Second paragraph. The 4 FGs are mentioned again, but later only FG1 and FG2 are mentioned, and without clarifying whether they are workshops or not. This doubt appears continuously to the reader throughout the paper.
11) Pag3. Second paragraph……. which applies Bourdieu’s theory on the (three ?) forms of capital to understand values in the SOI in the agri-food value-chain. Missing quotation.
12) Pag3. 2. Values and SOI There is another part with the same numbering. 2 Material and Methods, then it seems that Values and SOI should go in the introduction point 1, or in point 2, but not as an independent point.
13) Pag4. Second paragraph. …….Values are paramount in the agri-food sector too [21][26][30][31][32][33][34][35][36][37][38][39][40][41][42][43]. 16 quotes for a 6-word sentence is something completely excessive and makes no sense.
14) Pag4. Third paragraph. reinforce quote 38, put some more.
15) Pag4. Third paragraph. ……transformative systemic change [45][46][47][48][49][50][51]. Misquoted as in much of the paper.
16) Pag 4. Last paragraph. The paper tries to have representation from Europe, but there are really no participants from e.g. Germany or Poland, very representative countries in Europe. This element should be put as a limitation of the paper in point 5 of Discussion.
17) Pag 4. Last line. Here we are talking about appendix A where the participants in the FGs are listed, but when looking at appendix A, one can see that the title of the appendix is Workshop participants. The authors should clarify this circumstance, as it makes it difficult to understand the methodology they have followed.
18) Pag 5. In relation to the previous point, it is understood that the workshop was for the RWG, but everything is extremely confusing.
19) Pag 5. If the first was the workshop, then those who participated in it are the ones in appendix A? Please clarify.
20) Pag 5 and 6. In the FGs there is only one consumer representative (Appendix A), so the authors, faced with this lack of representation, add 53 consumers forming two new groups of FGs? And the total is 4? Why have these 53 consumers not been included in the table of Appendix A? From which country are these consumers? If these consumers generated 2 new FGs, without the presence of the other actors in the supply chain, how could this circumstance affect the research bias?
21) Pag 6. First paragraph. …real-life situations [52][53][54]. Misquote
22) Pag 6. Last paragraph. …… each one of them representing a character in the agri-food value-chain (food retailer, farmer, tourism operator, etc.). It is inconsistent with the presentation of the results when it is said that the supply chain actors are consumers, retailers, producers, authorities, and advisors.
23) Pag 7. Results. The results are presented organised according to the two workshops, FG1 and FG2; the former, based on the use of vignettes and brainstorming, the latter based on storyboarding and building from the previous workshop. Again, mixing workshops and focus groups. There is also a line width change in the text.
24) Pag 7. Punt 3.1. Actors in the supply chain, it is necessary to justify why these actors were chosen and quote previous work, and if there is no previous work, justify it adequately.
25) Pag 7 and 8. 3.1.1. to 3.1.5. You should follow the same order in the presentation of the results and in the same way, you do it in the last three but not in the first two. The authors should consider presenting the results in form of a table, which would be better for the reader.
26) 3.2. FG2. Reference is made here to appendix C. In the revision of appendix C, the number of groups is differentiated in parentheses, but there are only 4 groups. But there were not 5 actors? What are these groups? Are there only the 4FGs? The authors should clarify all this. I can not understand the methodology and the results.
27) In appendix B, there are not 5 actors, instead we find 4 actors: Government, Advisors, Producer (big and small), and Retailer. Consumers do not appear, and producers have been separated into two subgroups. The separation occurs without having been considered or named in the text. It is again very difficult to follow and understand.
28) Pag 9. Discussion. …. Change of line width?
29) Pag 9. Discussion. …. Their decision-making mechanisms are far more complex than a mere push towards maximisation. Add economic.
30) Pag 10. Second paragraph. Misquote [45]…..
31) Pag 10. Second paragraph. However, social capital within that net-work can act both as an enabler, but also as a hinderer of SOI. Please, explain why.
32) Pag 10. Third paragraph. Misquote
33) References. The number of citations used, more than 80, seems completely out of place. I think the authors should seriously consider reducing them. It is also noted that there is a high number of citations that are very old.
Author Response
Dear reviewer 1, we are extremely grateful of your constructive, engaged and useful feedback. We feel that the revised version, while addressing your concerns, is a much-improved version of the original submission, of which we are very thankful.
We will address your feedback one by one:
0) The word workshop has been avoided in the appendix and whenever talking about the FGs, and the wording was changed both in the abstract and the introduction, to clarify the fact that the four FGs are really two sessions of two groups. This was also further emphasized in the methods section. We hope this clarifies the issue.
1) This misquotes have been solved.
2) See 3)
3) The indentation on the second paragraph has been changed to make it even clearer that it is a textual quote. Hope this clarifies it. Because it is an extended quote (more than 40 words) a different indentation and separate paragraph is used, instead of quotation marks.
4) A working definition of SOI was introduced in paragraph 4, when the concept is first introduced.
5) There is no author’s guides on this in particular, but it is often the case that concepts are initialed in the abstract and the main body of the text separately. This is our experience, but we are open to change it if the editors required this to be done.
6) This has been changed to “see [16,17]”
7) We are aware it is a very old quote, but it is an important old quote -a classic reference, so to speak- that shows that these debates are not new and have, indeed, been going on for some time.
8) Parenthesis has been closed, thanks for spotting this. References have also been changed.
9) Done, added a line explaining the link and pointing to the discussion in section 2.
10) This was addressed in 0)
11) It is explained in the next paragraph, but the reference has been added.
12) Thanks for pointing to the incorrect numbering of the sections, This has been corrected. Originally, values were part of the introduction (1), but we then decided to make it a separate point for that is the theoretical framework. Only after the theoretical framework we introduce the methods (3), so the numbering has been corrected throughout the paper.
13) We have reduced significantly the number of references to 53.
14) All the quotes at the end of the paragraph are reinforcing quote 38, which is but an example.
15) Ibid 13)
16) Inasmuch as we agree that both Poland and Germany are very important countries, we fail to see why not having them is a limitation. We never claimed that this was representative of the whole of Europe, but that there are representatives of across Europe, and that we needed to have our own sample (that is, participants in the FGs) to be representative of the countries and the regions represented in our project. We say that explicitly “the countries where the Ploutos project has partners”. Our research is therefore limited to those countries represented in the project, so we could not possibly include Germany or Poland. Still, 13 other European countries allow an important insight into the matter.
17) We called the FGs as workshops. This has been rectified for clarity’s sake and all mentions of workshops were changed for FGs, except those of the RWG.
18) See 17)
19) See 17)
20) We have changed the wording slightly to emphasise that the 53 participants represent a range of actors in the agri-food ecosystem; consumers were poorly represented, only by one participants (as seen in Table A). We hope the slightly modified wording avoids confusion in this respect.
21) Sorted.
22) We added a sentence on why we limited actors to five in FG1, and why we allowed participants to come up with more actors in FG2. We hope this clarifies this issue.
23) Solved.
24) Done, we have explained why we chose those actors.
25) Done. It is indicated in the text that tables with these results are in appendix B and appendix C, respectively.
26) The number in the parenthesis in the table were not necessary (they were used internally to disaggregate the data obtained for each one of the four storyboards created), so as they do not offer anything and can be confusing, we got rid of them. Hope this is now clearer.
27) Of course, this was the wrong table, and now I have included the correct one. Thank you very much for pointing to this.
28) No, it gives that impression because there are bullet points in previous paragraph, but width is consistent as far as we can see.
29) Added “economic” as suggested.
30) Solved.
31) Done, added a line to explain social capital as a hindrance to Soi in some occasions.
32) Solved.
33) We have reduced significantly the number of references to 53.
Thanks for spotting a number of issues we had overlooked; we hope that this version is acceptable for the reviewer. And once again, thank very much, the feedback received makes a much tighter paper and we are far happier with this version.
(comments attached in word document too)

Reviewer 2 Report
Currently, many of the works that are being carried out and published on issues as important as innovation or sustainability must bear in mind the message that this article conveys, such as the role that people play in contributing values , as well as involving all the stakeholders that make up the ecosystem to be analyzed, due to the wealth of values they provide.
It is an interesting work, supported by an EU project (H2020) and with very valuable information on the results of other projects. But it would be convenient to incorporate some more information on the methodology used in order to follow up on the research, for example, the use of vignettes is indicated (in the FG1 workshop) it would be useful to introduce a photo with the starting vignette, to know how after the Brainstorming that vignette has been enriched with the values contributed by the actors of the different links of the value chain. It would also be interesting to group these contributions by link and graphically represent the interferences that appear between them, in which it could be detected which are the links that support each other the most or have common interests.
On the other hand, in the second workshop, based on a Stormboarding exercise, as in the previous case, it is advisable to accompany photos to have an idea of the progress of the contributions. Pointing out among the contributions which has been, in the opinion of the enablers, the one that has contributed the most innovative elements and that help sustainability.
Author Response
Dear reviewer 2, thanks you very much for your helpful and constructive feedback. In relation to your observations, we have clarified in the revised version of the paper that grouping the contributions into categories was carried during the FGs -now we have added images of the tables with values and enablers/hindrances to SOI in the paper, so this will convey an idea of how this work was done collaboratively with participants during the workshops.
The vignettes have also been introduced as appendix D, and this will give a clearer idea of the work which was conducted.
We have also added, as requested, a few examples of the storyboards to give an idea of how they look like and how they can be used to discuss and negotiate values between different actors in the course of role-play. We hope this is much clearer now, in terms of the methods and how they were used.
Once again, many thanks for helping us improve this paper. (comments included as attachment too)

Reviewer 3 Report
The article is part of a project with partners from 13 coutries. There may be some ethical issues. The authors should explain why this is their work and that nobody else from the project had a significant contribution for this study.
In Introduction the authors state: "The Ploutos project focuses on rebalancing the value chain for the agri-food system, along three innovation streams: Behavioural Innovation; Sustainable Collaborative Business Model Innovation; and Data-driven Technology Innovation." They select Behavioural Innovation for this article. Before describing the theory, the authors should:
1. Describe the three innovation streams;
2. Explain why did they choose Behavioural Innovation.
Results should be related with the literature.
Contributions of the research, what does it bring new, implications for research and practice should be added.
Author Response
Dear reviewer 3, first of all we would like to thank you for your time and engagement with our paper, and for the most useful feedback. In relation to ethical issues, as a EU project, this project has ethical approval, as specified in the ethical considerations statement. In relation to why the authors took a lead in this paper, this is explained by the fact that we were the main coordinators of the WP on sustainability and behavioural innovation, and that it was our organisation, indeed, that was leading the particular task on discovering values involved in SOI. Although there are many partners involved in the project, few of them are engaged in research activities, and most of them are engaged in practical SOI. So, in this particular case, it was only the authors who were responsible for the execution of this research task.
We have explained in the revised paper that this is a mostly practical project, and what our role in the project was, which is also useful to positioning ourselves vis-à-vis our research.
We have also added a few words to explain better the three innovation streams, and we have also explained why we chose behavioural innovation (because that is precisely the stream in which we worked and were we led the task relevant to this paper).
Finally, we have re-worked the discussion and conclusion to make clearer both our contribution to the field, as well as the links between results and literature. We hope these changes satisfy the reviewer.
(comments attached in a word document too)

Round 2
Reviewer 1 Report
The authors in this new version have improved the gaps and questions raised by the previous version, achieving an article worthy of publication.
Sincerely